# Differentiation of *Leishmania*
*(L.)*
*infantum*, *Leishmania*
*(L.)*
*amazonensis* and *Leishmania*
*(L.)*
*mexicana* Using Sequential qPCR Assays and High-Resolution Melt Analysis

**DOI:** 10.3390/microorganisms8060818

**Published:** 2020-05-29

**Authors:** Marcello Ceccarelli, Aurora Diotallevi, Gloria Buffi, Mauro De Santi, Edith A. Fernández-Figueroa, Claudia Rangel-Escareño, Said A. Muñoz-Montero, Ingeborg Becker, Mauro Magnani, Luca Galluzzi

**Affiliations:** 1Department of Biomolecular Sciences, University of Urbino “Carlo Bo”, 61029 Urbino (PU), Italy; m.ceccarelli3@campus.uniurb.it (M.C.); aurora.diotallevi@uniurb.it (A.D.); g.buffi@campus.uniurb.it (G.B.); mauro.desanti@uniurb.it (M.D.S.); mauro.magnani@uniurb.it (M.M.); 2Department of Population Genomics, Computational and Integrative Genomics, National Institute of Genomic Medicine, Mexico City 14610, Mexico; efernandez@inmegen.gob.mx (E.A.F.-F.); crangel@inmegen.gob.mx (C.R.-E.); samunoz@inmegen.edu.mx (S.A.M.-M.); 3Centro de Medicina Tropical, Unidad de Investigación en Medicina Experimental, Facultad de Medicina, Universidad Nacional Autónoma de México, Hospital General de México, México City 04360, Mexico; becker@unam.mx; 4School of Engineering and Sciences, Tecnologico de Monterrey, Queretaro 76130, Mexico

**Keywords:** leishmania infantum, leishmania amazonensis, leishmania mexicana, qPCR, high resolution melting, kDNA, ITS1

## Abstract

*Leishmania* protozoa are the etiological agents of visceral, cutaneous and mucocutaneous leishmaniasis. In specific geographical regions, such as Latin America, several *Leishmania* species are endemic and simultaneously present; therefore, a diagnostic method for species discrimination is warranted. In this attempt, many qPCR-based assays have been developed. Recently, we have shown that *L. (L.) infantum* and *L. (L.) amazonensis* can be distinguished through the comparison of the Cq values from two qPCR assays (qPCR-ML and qPCR-ama), designed to amplify kDNA minicircle subclasses more represented in *L. (L.) infantum* and *L. (L.) amazonensis*, respectively. This paper describes the application of this approach to *L. (L.) mexicana* and introduces a new qPCR-ITS1 assay followed by high-resolution melt (HRM) analysis to differentiate this species from *L. (L.) amazonensis*. We show that *L. (L.) mexicana* can be distinguished from *L. (L.) infantum* using the same approach we had previously validated *for L. (L.) amazonensis*. Moreover, it was also possible to reliably discriminate *L. (L.) mexicana* from *L. (L.) amazonensis* by using qPCR-ITS1 followed by an HRM analysis. Therefore, a diagnostic algorithm based on sequential qPCR assays coupled with HRM analysis was established to identify/differentiate *L. (L.) infantum*, *L. (L.) amazonensis*, *L. (L.) mexicana* and *Viannia* subgenus. These findings update and extend previous data published by our research group, providing an additional diagnostic tool in endemic areas with co-existing species.

## 1. Introduction

Leishmaniasis is caused by many *Leishmania* species belonging to subgenera *Leishmania* (*Leishmania*) and *Leishmania* (*Viannia*), creating a global public health problem with 0.2–0.4 million cases of visceral leishmaniasis (VL) and 0.7–1.2 million cases of cutaneous leishmaniasis (CL) per year [1]. In specific geographical regions, such as Central and South America, many *Leishmania* species are endemic and simultaneously present and, in some cases, can give rise to the same clinical form [2]. For instance, *L. (L.) amazonensis* and *L. (L.) mexicana* are responsible for the cutaneous manifestations and have a wide geographic distribution from Mexico to the north of Argentina. The epidemiological heterogeneity and difficulties in clinical approaches make species identification a critical step in clinical diagnosis and management, especially in case of co-infections. Therefore, an accurate diagnostic method allowing species identification is necessary [3]. Since the microscopic analysis does not provide information for species discrimination and the isoenzymatic characterization (i.e., multilocus enzyme electrophoresis) is a challenging and time-consuming technique, many biomolecular assays have been developed for *Leishmania* detection and species identification [4,5]. In particular, many qPCR assays have been designed to target the ribosomal DNA (rDNA) sequences and the kinetoplast DNA (kDNA) minicircle network that characterizes the *Leishmania* genus [6,7]. The rDNA sequence is repeated tens or hundreds of times per cell, allowing acceptable sensitivity also with DNA from clinical samples. Moreover, the variability of the internal transcribed spacer (ITS) sequences can be exploited for typing at the species level [8]. The kDNA minicircles are about 800 bp long and are present in several thousands of copies per cell, which makes them ideal targets for highly sensitive PCR-based assays. Minicircles are characterized by conserved regions in their replication origin [9], which allows the design of PCR primers with broad taxonomic coverage. On the other hand, the minicircle network is composed of different subclasses presenting high sequence variability, with exception of the conserved regions [10]. The number and identity of minicircle subclasses vary greatly among *Leishmania* species [11,12,13]. These features make the design of qPCR assays for identification at the species level difficult to perform, even using probes or melting analysis [14,15,16,17]. To differentiate *L. (L.) infantum* from *L. (L.) amazonensis*, we recently proposed an approach based on the evaluation of relative abundance of minicircle subclasses by using two qPCR assays [18,19,20]. However, our previous works did not include *L. (L.) mexicana*, which is closely related to *L. (L.) amazonensis* [21].

In this paper, we also applied this diagnostic approach to *L. (L.) mexicana* species, with the aim to extend the validity of our previous work. Results showed that, as previously demonstrated for *L. (L.) amazonensis*, *L. (L.) mexicana* can also be distinguished from *L. (L.) infantum* by exploiting the two qPCR assays designed on minicircle kDNA. Moreover, it was also possible to reliably distinguish *L. (L.) mexicana* from *L. (L.) amazonensis* species using a new high-resolution melt (HRM) assay designed on the ITS1 region (qPCR-ITS1).

## 2. Materials and Methods

### 2.1. Leishmania Strains, Clinical Samples and DNA Extraction

The *Leishmania* strains and isolates used in this study are listed in Table 1. The *L*. *(L.)*
*mexicana* clinical isolates 2, 3 and 5 were from diffuse cutaneous leishmaniasis lesions, whereas the clinical isolates 14, 17 and MHOM/MX/2011/Lacandona were from localized cutaneous leishmaniasis lesions. The clinical samples were taken from individuals from Quintana Roo, an endemic area of leishmaniasis in Mexico, as well as from patients treated at the Tropical Medicine Center, Medical Faculty, National Autonomous University of Mexico (UNAM); all patients were clinically diagnosed as diffuse or localized cutaneous leishmaniasis by Giemsa-stained smears of the lesions and by ELISA test for *Leishmania* (Table 2).

The DNA was extracted from promastigote cultures and from clinical samples using phenol-chloroform standard procedures followed by ethanol precipitation and the High Pure PCR Template Preparation Kit (Roche, Mannheim, Germany). The DNA was quantified using a Qubit fluorometer (Life Technologies, Carlsbad, USA) and stored at −20 °C until being used.

### 2.2. ITS1-PCR RFLP

The *L*. *(L.)*
*mexicana* strains and clinical samples were typed using ITS1-PCR RFLP according to Monroy-Ostria et al. [22]. Briefly, the PCR was performed using primers LITSR and L5.8S, following the amplification protocol—94 °C for 4 min followed by 36 cycles of 94 °C for 40 s, 54 °C for 30 s and 72 °C for 1 min and a final extension at 72 °C for 6 min. PCR products were nested using the same PCR conditions for 18 cycles. PCR products were digested with *Hae*III for 3 h at 37 °C and for 20 min at 80 °C to inactivate the enzyme. The restriction fragments were subjected to electrophoresis on a 4% agarose gel.

### 2.3. DNA Sequencing and Phylogenetic Analysis

The alanine aminotransferase (*ALAT*) gene was amplified in clinical samples px2, px3, px9, px10, pxJLC and in *L.* (*L.*) *mexicana* MHOM/MX/2011/Lacandona according to Marco et al. [23] using primers ALAT.F and ALAT.R. The amplification conditions were—94 °C for 3 min followed by 40 cycles of 94 °C for 30 s, 55 °C for 30 s and 72 °C for 30 s. PCR products were purified using the Agencourt AMPure XP kit (Beckman Coulter, Brea, CA, USA) and sequenced using the BigDye Terminator v3.1 Cycle Sequencing Kit (Thermo Fisher Scientific, Waltham, MA, USA) on ABI 3730xL DNA Analyzer (Thermo Fisher Scientific, Waltham, MA, USA). Chromatograms were visualized with ApE software and consensus sequences were generated and compared between them and with other validated species of *L. (L.) mexicana* deposited in GenBank using the Blastn tool available in the same platform. A phylogenetic reconstruction based on the Maximum Likelihood (ML) method was generated and a phylogenetic tree was constructed with 10,000 bootstrap replications, using the close-neighbor interchange method in Mega 6.0.

### 2.4. qPCR Assays

The qPCR-ML (amplifying kDNA minicircle subclass more represented in *L.* (*L.*) *infantum*) and qPCR-ama (amplifying kDNA minicircle subclass more represented in *L*. *(L.)*
*amazonensis*) were performed as previously described [18]. The new assay qPCR-ITS1 was performed using the new primers ITS1mexama_F (5′-GGATCATTTTCCGATGATTACACC-3′) and ITS1mexama_R (5′-CTGCAAATGTTGTTTTTGAGTACA-3′), flanking a portion of ITS1 sequence containing differences between *L*. *(L.)*
*amazonensis* and *L.* (*L.*) *mexicana* (Figure 1). The primers were designed using Primer BLAST and were verified against the ITS1 sequences of *L*. *(L.)*
*amazonensis* (*n* = 32) and *L.* (*L.*) *mexicana* (*n* = 30) encompassing forward and/or reverse primers, available in the Genbank database.

For all assays, PCR reactions were carried out in a 25 μL volume with 1–3 μL of template DNA using SYBR green PCR master mix (Diatheva srl, Fano, Italy) or TB Green premix ex TaqII Mastermix (Takara Bio Europe, France) and 200 nM of each primer in a Rotor-Gene 6000 instrument (Corbett Life Science, Mortlake, Australia). The amplification conditions were—94 °C for 10 min, 40 cycles at 94 °C for 25 s, 60 °C for 20 s and 72 °C for 20 s. At the end of each run, a melting curve analysis was performed from 78 to 92 °C with a slope of 1 °C/s, and 5 s at each temperature. The reactions were performed in duplicate. Dilution curves (from 1.0 to 1 × 10^−5^ ng/reaction) were established using *L.* (*L.*) *mexicana* MHOM/MX/2011/Lacandona DNA for qPCR-ML, qPCR-ama and qPCR-ITS1. The threshold cycles (Cq) were determined using the quantitation analysis of the Rotor-Gene 6000 software, setting a threshold to 0.15. To evaluate the potential interference of host DNA as a background in the qPCR analysis, 30 ng of human DNA was spiked in the reaction tubes.

### 2.5. High-Resolution Melt (HRM) Analysis

The qPCR-ML, qPCR-ama and qPCR-ITS1 amplicons were analyzed by HRM protocol on a Rotor-Gene 6000 instrument as described previously [24] with few modifications. Briefly, HRM was carried out over the range from 80 to 90 °C (qPCR-ML, qPCR-ama) or 75 to 85 °C (qPCR-ITS1), rising at 0.1 °C/s and waiting for 2 s at each temperature. Each sample was run in duplicate, and the gain was optimized before melting on all tubes.

### 2.6. Ethics Approval

This study was conducted according to the principles expressed in the Declaration of Helsinki. This research was approved by the Institutional Ethics Committee of the Medical Faculty of the National Autonomous University of Mexico (FM/DI/013/SR/2019). Guidelines established by the Mexican Health Authorities were strictly followed. All patients received treatment and clinical care by health authorities and signed a written informed consent for the collection of samples and subsequent analysis.

### 2.7. Statistical Analysis

Statistical analysis was performed with GraphPad Prism 5.0 (GraphPad Software, San Diego, CA, USA). Normal distribution of data was assessed by D′Agostino and Pearson omnibus normality test (alpha = 0.05). Difference between Tm mean values was evaluated using the nonparametric Mann–Whitney test.

## 3. Results

### 3.1. Both L. (L.) mexicana and L. (L.) amazonensis Can be Distinguished from L. (L.) infantum Exploiting A Differential qPCR Targeting Minicircle kDNA

Previously, we have shown that *L*. *(L.)*
*infantum* and *L.*
*(L.)*
*amazonensis* can be distinguished by comparing the Cq values of two qPCR assays (qPCR-ML and qPCR ama). In this work, the qPCR-ML and qPCR-ama were sequentially performed using DNA from *L.* (*L.*) *mexicana* MHOM/MX/2011/Lacandona, isolate 2, isolate 3, isolate 5, isolate 14 and isolate 17 as templates. As already shown for *L.* (*L.*) *amazonensis* strains, Cq values obtained with qPCR-ama were much lower compared to those obtained with qPCR-ML (Table 3).

The Cq difference between qPCR-ama and qPCR-ML allowed us to include *L.* (*L.*) *mexicana* among the *Leishmania* (*Leishmania*) species that can be distinguished from *L*. *(L.)*
*infantum.* Results from *L.*
*(L.)*
*amazonensis* MHOM/BR/00/LTB0016 and *L*. *(L.)*
*infantum* MHOM/FR/78/LEM75 were included as representative results obtained previously. As a consequence of different minicircle subclass amplified, the qPCR-ML and qPCR-ama showed a different limit of detection, allowing to amplify up to 1.0 × 10^−2^ and 1.0 × 10^−5^ ng of *L.* (*L.*) *mexicana* MHOM/MX/2011/Lacandona DNA, respectively (Table 3). In the qPCR-ML, the presence of 30 ng of purified human DNA delayed the limit of detection to 1.0 × 10^−1^ ng (Appendix A). With regard to qPCR-ama, the efficiency and detection limit were evaluated using 10-fold *L.* (*L.*) *mexicana* MHOM/MX/2011/Lacandona DNA serial dilutions (from 1.0 to 1×10^−5^ ng) in three independent experiments. There was a linear correlation between the log of DNA concentration and Cq value (slope = −3.3909, R^2^ = 0.9716) with a reaction efficiency of 97%. In order to evaluate the interference of host DNA, the DNA dilutions were spiked with 30 ng of purified human DNA, showing a delay on the Cq values but with comparable efficiency and limit of detection (Figure 2). The efficiency and detection limit obtained with *L.* (*L.*) *mexicana* DNA were in agreement with previous results obtained using DNA template from *L*. *(L.)*
*amazonensis* [18].

The qPCR-ML/qPCR-ama approach was also applied to 11 clinical samples. These samples were characterized as *L.* (*L.*) *mexicana* by ITS1-PCR RFLP (Appendix A), with the exception of pxCMU, for which a digestion profile could not be obtained. Moreover, the genotype of five clinical samples (px2, px3, px9, px10, pxJLC) were further confirmed as *L.* (*L.*) *mexicana* by sequencing and phylogenetic analysis of the alanine aminotransferase (ALAT) gene (Appendix A). All samples showed Cq qPCR-ama <Cq qPCR-ML (Table 4), confirming the presence of *L.* (*L.*) *mexicana* parasites.

### 3.2. L. (L.) amazonensis Can be Differentiated from L. (L.) mexicana by qPCR-ITS1 HRM Analysis

In the attempt to differentiate *L.* (*L.*) *mexicana* and *L.* (*L.*) *amazonensis*, HRM analyses were performed after qPCR-ML and qPCR-ama using the *L.* (*L.*) *mexicana* and *L.* (*L.*) *amazonensis* samples indicated in Table 1 and Table 2. However, both assays did not allow us to distinguish the two species reliably. In particular, the qPCR-ML assay showed overly high Cq values (>30) in *L.*
*(L.)*
*mexicana* samples. Concerning the qPCR-ama assay, HRM analysis of all *L.* (*L.*) *mexicana* and *L.* (*L.*) *amazonensis* samples showed heterogeneous profiles (Appendix A). Moreover, despite that the mean Tm of PCR products from *L.* (*L.*) *mexicana* and *L.* (*L.*) *amazonensis* were significantly different (Mann–Whitney test, *p* < 0.01), the Tm value distributions partly overlapped, de facto making the distinction between the two species unreliable (Figure 3).

Therefore, a new qPCR assay and HRM analysis were designed on ITS1 sequences. The in silico analysis showed that PCR product lengths were 125–126 and 129–131 bp for *L.* (*L.*) *amazonensis* and *L.* (*L.*) *mexicana*, respectively. The qPCR-ITS1 efficiency and detection limit were evaluated using 10-fold *L*. *(L.)*
*mexicana* MHOM/MX/2011/Lacandona DNA serial dilutions (from 1.0 to 1 × 10^−4^ ng). A linear correlation between the log of DNA concentration and Cq value was demonstrated (slope = −3.6227, R^2^ = 0.997), with a reaction efficiency of 89%. As shown for the qPCR-ama, spiking with 30 ng of purified human DNA induced a delay on the Cq values, but efficiency and limit of detection were not affected (Figure 4). The Tm analysis of qPCR-ITS1 amplicons obtained from all amplified *L.* (*L.*) *amazonensis* and *L.* (*L.*) *mexicana* samples allowed full discrimination between the two species (Mann–Whitney test, *p* < 0.001) (Figure 5) (Appendix A). However, three clinical samples failed to amplify (Px7, PxGSF, PxCMU). Overall, the qPCR-ITS1 HRM assay for *amazonensis*/*mexicana* species discrimination showed 84.2% sensitivity and 100% specificity.

## 4. Discussion

The identification of *Leishmania* species is an important diagnostic aspect, especially in Latin America, not only for epidemiological studies but also for the accurate monitoring of clinical disease evolution. In fact, the only species causing VL in this geographical region is *L*. *(L.)*
*infantum* (syn. *chagasi*), while cutaneous or mucocutaneous (MCL) manifestations can also be generated by *Viannia* subgenus and *L.*
*(L.)*
*mexicana* complex. In this epidemiological and clinical context, the species discrimination appears pivotal, e.g., to monitor a cutaneous lesion that could evolve in VL, MCL or disseminated CL, depending on the species. In this view, molecular diagnostic tools allowing species discrimination can be helpful. The kDNA minicircles are ideal targets for highly sensitive molecular detection of *Leishmania* spp. since they are present in thousands of copies per cell [25]. Since the pioneering work of Nicolas et al. [26], many qPCR assays have been designed on conserved regions of minicircles to detect/quantify *Leishmania* parasites. Moreover, several authors investigated the possibility to exploit minicircle sequences to discriminate *Leishmania* parasites at the species level, reaching only partial results due to the variability of minicircle subclasses [15,16]. Previously, we proposed an SYBR Green qPCR-based approach to distinguish *L*. *(L.)*
*infantum* from *L*. *(L.)*
*amazonensis*, exploiting the different abundance of minicircle subclasses rather than targeting a species-specific sequence. Using this approach, which relies on two qPCR assays (qPCR-ML and qPCR-ama) and evaluation of Cq values, we were able to distinguish the two species adequately [18].

In this work, we tested this approach with *L.* (*L.*) *mexicana*, which is genetically close to *L.*
*(L.)*
*amazonensis*. The comparison of Cq values of qPCR-ML and qPCR-ama confirmed results previously obtained with *L*. *(L.)*
*amazonensis,* allowing us to include *L.* (*L.*) *mexicana* among the *Leishmania* (*Leishmania*) species that can be differentiated from *L.* (*L.*) *infantum*, therefore extending the conclusion of our previous work. Importantly, this approach was successfully applied to cutaneous lesions of 11 patients diagnosed with diffuse or localized cutaneous leishmaniasis. Notably, the clinical sample pxCMU, which was negative in ITS1-PCR RFLP, was identified as *L*. *(L.)*
*mexicana*/*amazonensis*, evidencing the highest sensitivity of our qPCR assays targeting minicircles. These results further support the possibility of exploiting the relative abundance of minicircles for *Leishmania* species discrimination. Moreover, we confirmed the use of an adequate diagnostic approach based on consecutive qPCR assays to define species [18], as also proposed by other authors [27].

The distinction between *L.*
*(L.)*
*amazonensis* and *L. (L.) mexicana* is important for epidemiological studies and disease monitoring, but it can be challenging [28]. For instance, *hsp70* analysis by Fraga et al. [29] did not resolve between these species. On the other hand, other authors were able to separate these species based on multilocus sequence typing (MLST) [30] or sequential real-time PCR assays [27].

The qPCR coupled with HRM analysis is considered as a good option in molecular diagnostics since it avoids the use of modified oligonucleotides, it is accurate, allows high-throughput applications and is faster and cheaper than other types of analysis such as MLST, RFLP or single-gene DNA sequencing. Moreover, since the qPCR is a closed-tube system, the potential for carryover contamination will be reduced. In the attempt to discriminate between *L*. *(L.)*
*amazonensis* and *L.* (*L.*) *mexicana*, HRM profiles of amplicons from qPCR-ama were investigated; however, their heterogeneity did not us allow to distinguish these two species reliably. Since Schönian et al. demonstrated the possibility to discriminate the two species using ITS1-PCR RFLP [31], we designed an HRM-based assay exploiting differences in *L*. *(L.)*
*amazonensis* and *L.* (*L.*) *mexicana* ITS1 sequences, in order to avoid restriction digestion and electrophoretic analysis. This process allows saving a considerable amount of time to perform the analysis and avoids possible difficulties in restriction fragment identification. As expected from the in silico sequence analysis, the observed HRM Tm values of all *L.* (*L.*) *mexicana* samples were significantly higher as those of all *L*. *(L.)*
*amazonensis* samples, allowing a robust distinction between these two species. The fact that three clinical samples did not amplify (Px7, PxGSF, PxCMU) was probably due to the lower sensitivity of qPCR-ITS1 as compared to the assay targeting kDNA minicircles.

## 5. Conclusions

In the attempt to use a qPCR-based approach to differentiate *Leishmania* species co-existing in the New World, sequential qPCR assays and HRM analysis have been implemented. The results showed that—(i) *L.*
*(L.)*
*infantum* can be distinguished from *L.* (*L.*) *mexicana* comparing the Cq values of qPCR-ML and qPCR-ama, as previously shown for *L.* (*L.*) *amazonensis*; (ii) this distinction was possible not only using strains/isolates but also in clinical samples; (iii) the differentiation between *L.* (*L.*) *amazonensis* and *L.* (*L.*) *mexicana* was achieved by qPCR-ITS1 HRM analysis. Therefore, it was possible to design/update an algorithm that allows us to identify/differentiate *L*. *(L.)*
*infantum*, *L*. *(L.)*
*amazonensis*, *L.* (*L.*) *mexicana* and *Viannia* subgenus with sequential qPCR assays coupled with HRM analysis targeting minicircle kDNA and ITS1 sequence (Figure 6), which further extends our previous work.

## Figures and Tables

**Figure 1 microorganisms-08-00818-f001:**
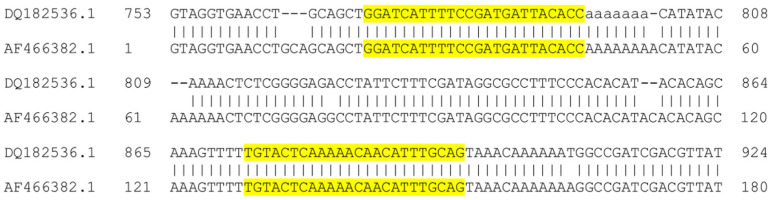
Alignment of ITS1 partial sequence of *L*. *(L.)*
*amazonensis* MHOM/BR/73/M2269 (acc. n. DQ182536.1) and *L*. *(L.)*
*mexicana* MHOM/MX/98/UNAM (acc. n. AF466382.1). Highlighted sequences represent primers ITS1mexama_F and ITS1mexama_R. Sequences are representative of *L*. *(L.)*
*amazonensis* (*n* = 32) and *L*. *(L.)*
*mexicana* (*n* = 30) ITS1 sequences available in the Genbank database.

**Figure 2 microorganisms-08-00818-f002:**
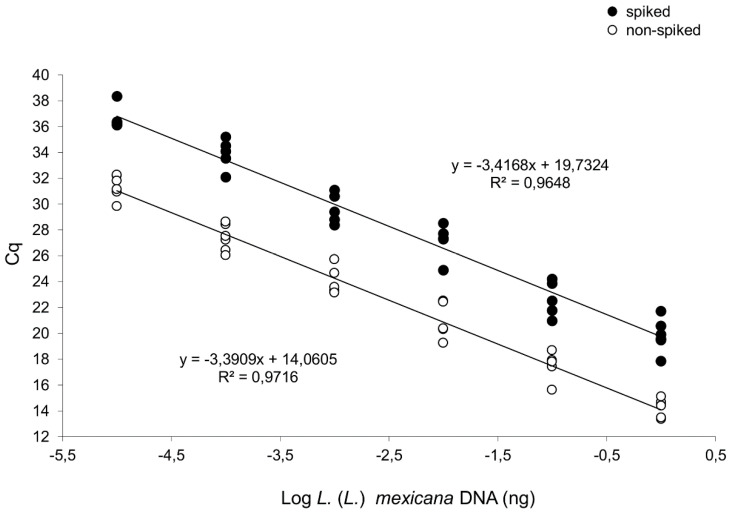
qPCR-ama curves constructed with serial dilutions of *L*. *(L.)*
*mexicana* MHOM/MX/2011/Lacandona DNA. The curves were obtained with serial dilutions ranging from 1.0 to 1.0 × 10^−5^ ng/tube, spiked with 30 ng human DNA (upper curve, y = −3.4168x + 19.73; R^2^ = 0.9648) or nonspiked (lower curve, y = −3.3909x + 14.0605; R^2^ = 0.9716). Results were from three independent experiments in duplicate.

**Figure 3 microorganisms-08-00818-f003:**
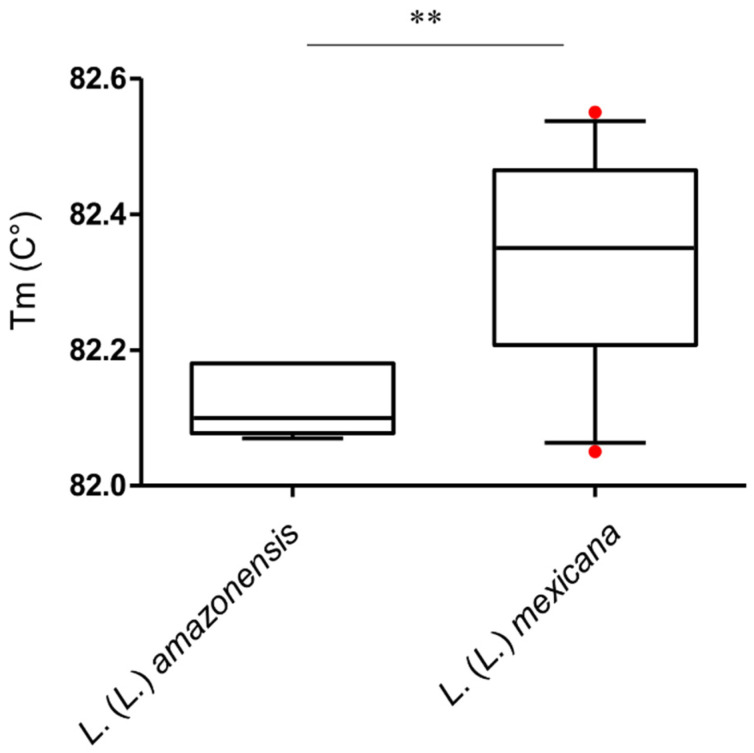
Box and whisker plot with 5–95% confidence interval showing Tm values distribution obtained with high-resolution melt (HRM) analysis of qPCR-ama amplicons of *L*. *(L.)*
*mexicana* (*n* = 32) and *L*. *(L.)*
*amazonensis* (*n* = 6). Line within the box represents the median and the red dots above and below the whiskers represent the outliers that are either greater than 95th or less than 5th percentile. ** *p* < 0.01.

**Figure 4 microorganisms-08-00818-f004:**
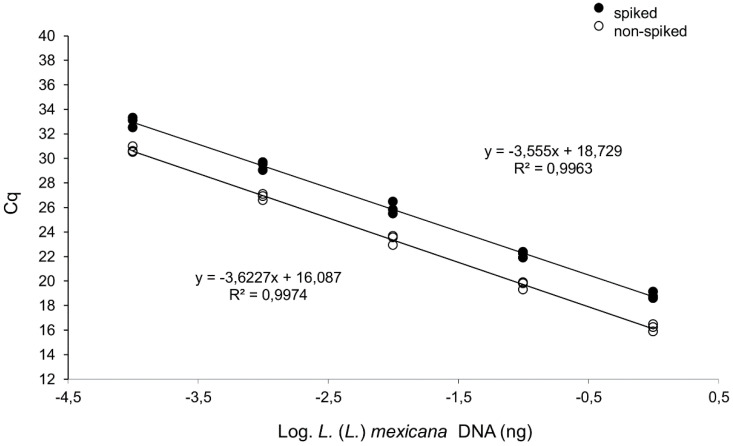
qPCR-ITS1 curves constructed with serial dilutions of *L*. *(L.)*
*mexicana* MHOM/MX/2011/Lacandona DNA. The curves were obtained with serial dilutions ranging from 1.0 to 1.0 × 10^−4^ ng/tube, spiked with 30 ng human DNA (upper curve, y = −3.555x + 18.729; R^2^ = 0.996) or nonspiked (lower curve, y = −3.623x + 16.087; R^2^ = 0.997). Triplicates of the PCR amplification are represented.

**Figure 5 microorganisms-08-00818-f005:**
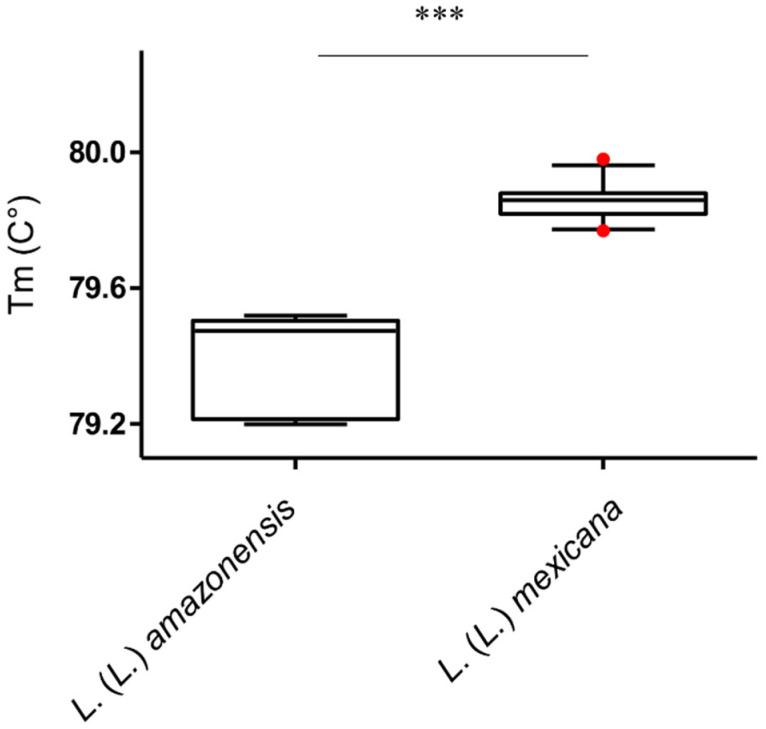
Box and whisker plot with 5–95% confidence interval showing Tm value distribution obtained with HRM analysis of qPCR-ITS1 amplicons of *L*. *(L.)*
*mexicana* (*n* = 26) and *L*. *(L.)*
*amazonensis* (*n* = 6). Line within the box represents the median and the red dots above and below the whiskers represent the outliers that are either greater than 95th or less than 5th percentile. *** *p* < 0.001.

**Figure 6 microorganisms-08-00818-f006:**
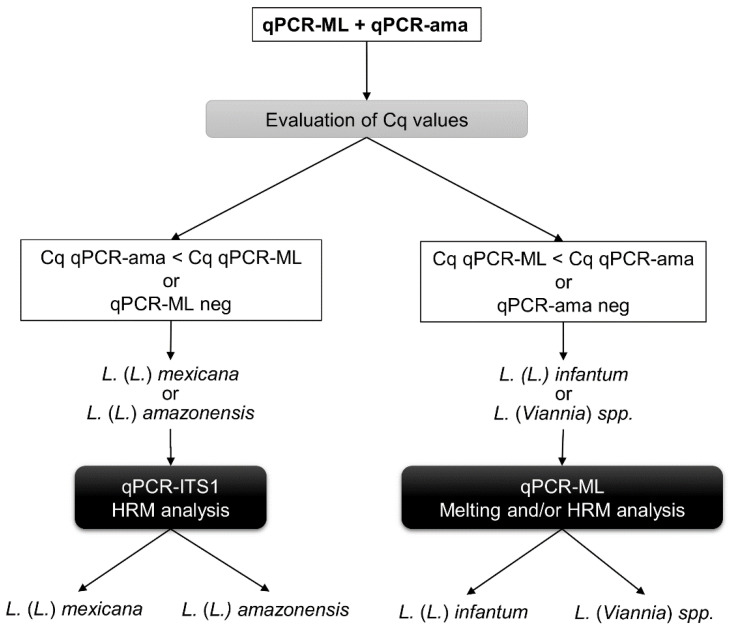
Updated sequential qPCR and HRM approach targeting minicircle kDNA and ITS1 sequence for the identification of *L*. *(L.)*
*infantum*, *L*. *(L.)*
*amazonensis*, *L.* (*L.*) *mexicana* and *Viannia* subgenus. First, the qPCR-ML and qPCR-ama followed by HRM analysis are performed. The evaluation of Cq values and HRM profiles for both assays will allow discriminating among *L*. (*Viannia*) spp., *L*. *(L.)*
*infantum* and the two species *L.*
*(L.)*
*mexicana/amazonensis*. Then, qPCR-ITS1 HRM analysis is performed to discriminate between *L*. *(L.)*
*amazonensis* and *L.* (*L.*) *mexicana*.

**Table 1 microorganisms-08-00818-t001:** *Leishmania* spp. reference strains/clinical isolates used in this study.

Species	Strain/Isolate
*L.* (*L.*) *infantum*	MHOM/TN/80/IPT1
*L. (L.) infantum*	MHOM/IT/86/ISS218
*L. (L.) infantum*	MHOM/FR/78/LEM75
*L. (L.) amazonensis*	MHOM/BR/00/LTB0016
*L. (L.) amazonensis*	IFLA/BR/67/PH8
*L. (L.) amazonensis*	Clinical isolate
*L. (V.) braziliensis*	MHOM/BR/75/M2904
*L. (L.) mexicana*	MHOM/MX/2011/Lacandona
*L. (L.) mexicana*	Clinical isolate 2
*L. (L.) mexicana*	Clinical isolate 3
*L. (L.) mexicana*	Clinical isolate 5
*L. (L.) mexicana*	Clinical isolate 14
*L. (L.) mexicana*	Clinical isolate 17

**Table 2 microorganisms-08-00818-t002:** Clinical samples used in this study.

Sample	Species Identification (ITS1-PCR RFLP)	Species Identification (*ALAT* sequencing)	CL Form
Px1	*L. (L.) mexicana*	n.a. ^1^	LCL
Px2	*L. (L.) mexicana*	*L. (L.) mexicana*	LCL
Px3	*L. (L.) mexicana*	*L. (L.) mexicana*	LCL
Px4	*L. (L.) mexicana*	n.a. ^1^	LCL
Px5	*L. (L.) mexicana*	n.a. ^1^	DCL
Px7	*L. (L.) mexicana*	n.a. ^1^	LCL
Px9	*L. (L.) mexicana*	*L. (L.) mexicana*	DCL
Px10	*L. (L.) mexicana*	*L. (L.) mexicana*	LCL
PxGSF	*L. (L.) mexicana*	n.a. ^1^	LCL
PxCMU	n.a. ^1^	n.a. ^1^	n.a. ^1^
PxJLC	*L. (L.) mexicana*	*L. (L.) mexicana*	LCL

^1^ not available.

**Table 3 microorganisms-08-00818-t003:** qPCR-ML and qPCR-ama results in strains/clinical isolates.

*Leishmania* Species, Strain/Isolate	DNA Template (ng)	qPCR-ML (Cq ± SD)	qPCR-ama (Cq ± SD)
*L*. *(L.)**mexicana* MHOM/MX/2011/Lacandona	1.0	31.61 ± 2.03	14.25 ± 0.69
*L*. *(L.)**mexicana* MHOM/MX/2011/Lacandona	1.0 × 10^−1^	33.43 ± 2.09	17.48 ± 1.13
*L*. *(L.)**mexicana* MHOM/MX/2011/Lacandona	1.0 × 10^−2^	37.53 ± 1.27	20.68 ± 1.46
*L*. *(L.)**mexicana* MHOM/MX/2011/Lacandona	1.0 × 10^−3^	n.d. ^1^	24.07 ± 1.09
*L*. *(L.)**mexicana* MHOM/MX/2011/Lacandona	1.0 × 10^−4^	n.d. ^1^	27.37 ± 1.05
*L*. *(L.)**mexicana* MHOM/MX/2011/Lacandona	1.0 × 10^−5^	n.d.^1^	31.36 ± 0.93
*L*. *(L.)**mexicana* Isolate 2	1.0	33.19 ± 1.34	16.78 ± 0.06
*L*. *(L.)**mexicana* Isolate 3	1.0	33.62 ± 2.14	18.62 ± 1.12
*L*. *(L.)**mexicana* Isolate 5	1.0	38.19 ± 1.01	20.14 ± 0.43
*L*. *(L.)**mexicana* Isolate 14	1.0	34.59 ± 0.51	16.54 ± 0.17
*L*. *(L.)**mexicana* Isolate 17	1.0	35.94 ± 1.20	19.15 ± 1.00
*L*. *(L.)**amazonensis* MHOM/BR/00/LTB0016	1.0 × 10^−1^	33.95 ± 0.34	21.1 ±1.02
*L*. *(L.)**infantum* MHOM/FR/78/LEM75	1.0	14.42 ± 0.75	28.02 ± 0.98

^1^ not detectable.

**Table 4 microorganisms-08-00818-t004:** qPCR-ML and qPCR-ama results in clinical samples.

Sample ID	qPCR-ML (Cq ± SD)	qPCR-ama (Cq ± SD)
Px1	n.d. ^1^	27.71 ± 0.08
Px2	n.d. ^1^	25.79 ± 0.55
Px3	n.d. ^1^	24.57 ± 0.58
Px4	36.21 ± 1.99	29.59 ± 0.80
Px5	n.d. ^1^	28.99 ± 1.57
Px7	35.31 ± 1.47	28.04 ± 0.25
Px9	n.d. ^1^	28.35 ± 1.93
Px10	n.d. ^1^	24.92 ± 0.87
PxGSF	36.38 ± 1.70	29.49 ± 0.53
PxCMU	36.53 ± 1.36	31.48 ± 0.36
PxJLC	n.d. ^1^	34.87 ± 1.93

^1^ not detectable.

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
