# Peer review of "Differentiation of Leishmania (L.) infantum, Leishmania (L.) amazonensis and Leishmania (L.) mexicana Using Sequential qPCR Assays and High-Resolution Melt Analysis"

_microorganisms, 2020, doi:10.3390/microorganisms8060818_

Round 1

Reviewer 1 Report

"This fact allowed to include L. (L.) mexicana among the Leishmania "  This starts a paragraph, but it's not clear what the fact is.

This is not a great concluding sentence: increased sampling will be needed to 305 further strengthen the discrimination capacity of this assay 

There is nothing wrong with this work,  however I think it isn't relevant, and would need to be convinced. The qPCR methods (and equipment) are not trivial, and must be compared to the near-universal availability of regular PCR, gel electrophoresis and sequencing. While a large center might choose to use this papers methods, and have a qPCR machine dedicated to it, and new/smaller center should consider more easily accessible methods.

Thus I have give low scores for relevance, novelty, and interest. 

If an effort is made to compare these methods favorably related to other simpler methods, and suggest an important clinical setting for these methods, I might better understand the relevance of this work.

Author Response

Response to Reviewer 1 Comments

"This fact allowed to include L. (L.mexicana among the Leishmania "  This starts a paragraph, but it's not clear what the fact is.

REPLY: this has been better specified as suggested (line 185)

This is not a great concluding sentence: increased sampling will be needed to 305 further strengthen the discrimination capacity of this assay 

REPLY: We agree with the referee: we deleted the final sentence (line 311)

There is nothing wrong with this work,  however I think it isn't relevant, and would need to be convinced. The qPCR methods (and equipment) are not trivial, and must be compared to the near-universal availability of regular PCR, gel electrophoresis and sequencing. While a large center might choose to use this papers methods, and have a qPCR machine dedicated to it, and new/smaller center should consider more easily accessible methods.

Thus I have give low scores for relevance, novelty, and interest. 

If an effort is made to compare these methods favorably related to other simpler methods, and suggest an important clinical setting for these methods, I might better understand the relevance of this work.

REPLY: In the manuscript, the regular PCR coupled to restriction digestion and electrophoresis analysis (ITS1 PCR RFLP) and DNA sequencing were used as reference methods to type the clinical samples. The ITS1 PCR RFLP and DNA sequencing allowed to identify the clinical samples as L. mexicana; these results were in full agreement with our qPCR-based method. These data were described in the original version of the manuscript (lines 200-204 and Figure S2). Following reviewer’s suggestion, this part has been better pointed out by adding another supplementary figure representing ITS1 PCR RFLP analysis (Figure S2 in the revised version of the manuscript). Moreover, we pointed out that sample pxCMU was not amplified with regular ITS1 PCR but it was amplified with qPCR targeting minicircles, which allowed greater sensitivity (lines 201-202; 284-286).   

The advantages of qPCR/HRM-based approach vs methods based on regular PCR, electrophoresis and sequencing were briefly included in the original version of the manuscript (discussion, lines 292-294, 297-300). Due to referee observation, we extended and better pointed out this part (lines 296-300; 305-307).

Finally, it is true that regular PCR and gel electrophoresis are available in every molecular biology lab but qPCR machines are becoming more and more common (over 700 publications in Pubmed having Leishmania and qPCR or real-time PCR as main topic). Even taking into consideration the cost, a qPCR machine is less expensive than small Sanger sequencing instrument (which costs over 50,000 $). Furthermore, PCR-RFLP often requires low-melt (high resolution) agarose which is usually very expensive. Therefore, despite the real-time PCR technology is not used yet for routine clinical application in low-income countries, it holds big potential for the near future.

Reviewer 2 Report

This manuscript presents new data on a potential diagnostic tool that will allow differentiation between three sympatric species of Leishmania, which constitutes one of the most important disease complexes in many countries.  The results will be very interesting to a wide range of scientists and diagnosticians.  The manuscript is very well written and organized.  The methods are sound and well explained.  The Materials and Methods are sound and are sufficient to allow falsification or corroboration.  The conclusions are consistent with the results.

It is rare for me to suggest no revisions, but in this case I believe this excellent study and manuscript should be published with no further revision.  Minor copy editing might be helpful, but even this is not necessary.  Given the large and growing importance of leishmaniasis in terms of global health, I recommend that this paper be published as quickly as possible.

Author Response

Response to Reviewer 2 Comments

This manuscript presents new data on a potential diagnostic tool that will allow differentiation between three sympatric species of Leishmania, which constitutes one of the most important disease complexes in many countries.  The results will be very interesting to a wide range of scientists and diagnosticians.  The manuscript is very well written and organized.  The methods are sound and well explained.  The Materials and Methods are sound and are sufficient to allow falsification or corroboration.  The conclusions are consistent with the results.

It is rare for me to suggest no revisions, but in this case I believe this excellent study and manuscript should be published with no further revision.  Minor copy editing might be helpful, but even this is not necessary.  Given the large and growing importance of leishmaniasis in terms of global health, I recommend that this paper be published as quickly as possible.

REPLY: We thank the referee for the very positive comments.